# Unraveling the Microbiota of Natural Black cv. Kalamata Fermented Olives through 16S and ITS Metataxonomic Analysis

**DOI:** 10.3390/microorganisms8050672

**Published:** 2020-05-06

**Authors:** Maria Kazou, Aikaterini Tzamourani, Efstathios Z. Panagou, Effie Tsakalidou

**Affiliations:** 1Laboratory of Dairy Research, Department of Food Science and Human Nutrition, School of Food and Nutritional Sciences, Agricultural University of Athens, Iera Odos 75, 11855 Athens, Greece; et@aua.gr; 2Laboratory of Food Microbiology and Biotechnology, Department of Food Science and Human Nutrition, School of Food and Nutritional Sciences, Agricultural University of Athens, Iera Odos 75, 11855 Athens, Greece; tzam.katerina@gmail.com

**Keywords:** table olives, Kalamata olives, Greek style fermentation, microbiological analysis, 16S metagenomics analysis, ITS metagenomics analysis, microbiota, high-throughput sequencing

## Abstract

Kalamata natural black olives are one of the most economically important Greek varieties. The microbial ecology of table olives is highly influenced by the co-existence of bacteria and yeasts/fungi, as well as the physicochemical parameters throughout the fermentation. Therefore, the aim of this study was the identification of bacterial and yeast/fungal microbiota of both olives and brines obtained from 29 cv. Kalamata olive samples industrially fermented in the two main producing geographical regions of Greece, namely Aitoloakarnania and Messinia/Lakonia. The potential microbial biogeography association between certain taxa and geographical area was also assessed. The dominant bacterial family identified in olive and brine samples from both regions was Lactobacillaceae, presenting, however, higher average abundances in the samples from Aitoloakarnania compared to Messinia/Lakonia. At the genus level, *Lactobacillus*, *Celerinatantimonas*, *Propionibacterium* and *Pseudomonas* were the most abundant. In addition, the yeasts/fungal communities were less diverse compared to those of bacteria, with Pichiaceae being the dominant family and *Pichia*, *Ogataea*, and *Saccharomyces* being the most abundant genera. To the best of our knowledge, this is the first report on the microbiota of both olives and brines of cv. Kalamata black olives fermented on an industrial scale between two geographical regions of Greece using metagenomics analysis.

## 1. Introduction

Table olives are one of the oldest and most popular fermented vegetables in the Mediterranean basin that has a significant contribution in the world production of fermented olives, with Spain, Greece, Italy and Portugal being among the main EU table olive producers (*ca.* 35% of world production). The renewed interest of consumers for high-quality food products with enhanced nutritional properties, such as bioactive compounds, dietary fibers, fatty acids and antioxidants, resulted in an increasing trend in the production of processed olives in recent years, which is expected to reach 2.9 million tons in the 2019/2020 season [1]. 

Table olive processing in Greece is of paramount socio-economic importance for the country. According to a recent report by the Greek Interprofessional Association for Table Olives [2], more than 64,000 farmers and 100 businesses are directly involved in the primary and secondary production and processing sectors, respectively. The annual amount of the end (processed) product has increased to 215,000 tons, from which 85% are exported, rendering the country as the second largest world exporter of table olives, right after Spain. The value of the market size of Greek table olive exports exceeds 450 million euros representing 9.2% of the overall exports of Greek agricultural commodities. Several trade preparations of Greek table olives are renowned brand names in the international market, including Halkidiki green olives, as well as Kalamata and Conservolea (or Amfissis) natural black olives. 

Although all trade preparations of table olives (i.e., treated, natural, dehydrated and/or shriveled olives, and olives darkened by oxidation) [3] are produced in the country, Greece has a long tradition in the production of natural black olives, which are internationally known as “*Greek style olives in brine*”, using olives from the two most economically important varieties, namely Conservolea and Kalamata [4]. In particular, the latter variety is highly esteemed for its exceptional organoleptic characteristics, the dark color and crisp texture of the end product increasing, therefore, the demand and the exports for Kalamata black olives. Nowadays, this variety is cultivated primarily in the prefectures of Aitoloakarnania (Western Greece), Messinia and Lakonia (Southern Peloponnese) and Fthiotida (Mainland Greece), but new olive trees are being planted every year in other areas as well. It is expected that, within the next 10 years, the production of Kalamata natural black olives will exceed 100,000 tons [2].

During Greek-style processing, olives are initially subjected to quality inspection to remove defective drupes. Subsequently, olives are immersed directly in a brine solution of about 6–10% (*w/v*) salt concentration where they undergo spontaneous fermentation for 8–12 months. The microbiota of table olives fermentation is a complex set of dynamics, in which the co-existence of lactic acid bacteria (LAB) and yeasts is of fundamental importance to obtain high quality products [5,6]. Until recently, the microbiota of table olive fermentation has been described mostly by culture-dependent approaches. However, it should be stressed out that almost 90% (or 99% according to other researchers) of microorganisms present on natural environments cannot be cultivated in synthetic microbiological media [7,8,9,10]. To overcome these limitations, culture-independent analyses and, most importantly, next generation sequencing (NGS) intensely changed the study of the microbial ecology of fermented foods [11,12,13]. Nowadays, high throughput sequencing (HTS) coupled with omics approaches are used for the characterization of food microbiota. Among them, amplicon-based metagenomics analysis is the most widely used. It should be noted that the majority of culture-independent approaches for the assessment of the microbial diversity in table olive fermentations have been applied on green olive fermentations, either by the Spanish method or directly brined [14,15,16,17,18,19,20,21,22,23,24,25]. However, limited information can be found in the literature for the microbial diversity of natural black olive fermentations using culture-independent approaches, especially for Greek table olive varieties [26]. 

The aim of this study was to unravel the microbial diversity of fermented natural black olives from the Kalamata variety using the state-of-the-art approach of amplicon-based metagenomics analysis. The samples were taken directly from fermentation vessels of processing installations located in different locations from two geographical regions in Greece, namely Aitoloakarnania and Messinia/Lakonia, where the main volume of this variety is cultivated and processed. To the best of our knowledge, this is the first study to elucidate the microbial community diversity of cv. Kalamata olives processed on industrial scale according to the traditional Greek-style method using the metagenomic approach.

## 2. Materials and Methods 

### 2.1. Origin of the Samples

Twenty-nine (29) samples of fermented cv. Kalamata (*Olea europea* var. *ceraticarpa*) natural black olives were obtained during the 2018–2019 season from two geographical areas in Greece, namely Aitoloakarnania in western Greece and Messinia/Lakonia in southern Peloponnese (Figure 1). The samples (3 kg each) were taken in coincidence with the final stage of black olive fermentation as traditionally implemented in the two regions. Overall, 14 samples were taken from the area of Aitoloakarnania and 15 samples from the area of Messinia/Lakonia (Table 1). Fermentation was undertaken in 5000-6000-L capacity polyester vessels containing 60–70% olives and 30–40% brine. Olives were processed according to traditional Greek-style anaerobic processing. Specifically, cv. Kalamata black olive processing included harvesting at the right stage of ripening (i.e., 3/4 of the mesocarp has attained black color), transportation to the processing installations, quality control inspection for the removal of defected drupes and finally brining in 5.0–6.0% (*w/v*) NaCl, where the olives were subjected to spontaneous fermentation by the autochthonous bacterial and fungal microbiota. During fermentation, salt level was adjusted to the initial concentration by periodic additions of coarse salt at the top of each fermentation vessel to facilitate the dominance of LAB over yeasts. In late May/early June, depending on ambient temperature, salt was added to the vessels to adjust salinity to 7.0–8.0% to avoid spoilage due to the upcoming high summer temperatures [27].

### 2.2. Microbiological Analysis

Classical microbiological analysis was performed in both olive and brine samples to enumerate the main microbial groups implicated in table olive fermentation, i.e., LAB, yeasts and enterobacteria [28]. For this purpose, olive (25 g) or brine (1 mL) samples were aseptically transferred into 225 mL or 9 mL sterile Ringer’s solution (0.9% (*w/v*) NaCl), respectively. Olive samples were homogenized in a stomacher device (LabBlender, Seward Medical, London, UK) for 60 s at room temperature. The resulting suspensions were serially diluted in the same diluent and duplicate 1 or 0.1 mL of the appropriate dilutions were poured or spread on the following agar media to enumerate: (i) LAB on de Man–Rogosa–Sharpe (MRS) medium (Biolife, Milan, Italy, pH adjusted to 5.7) containing 0.05% (w/v) cycloheximide (AppliChem GmbH, Darmstadt, Germany) and incubated at 25 °C for 72 h under anaerobic conditions (double agar layer); (ii) yeasts and molds on Rose Bengal Chloramphenicol agar (RBC; supplemented with selective supplement X009, Bury, UK), incubated for 48 h at 25 °C; and finally (iii) Enterobacteriaceae on Violet Red Bile Glucose agar (VRBGA; Biolife, Milan, Italy), incubated under anaerobic conditions (double agar layer) for 24 h at 37 °C. Thereafter, agar media containing 30–300 colonies were used for the enumeration of microbial counts and 20% of colonies present in the plates were randomly selected and examined for typical morphological traits related with each medium. The results were log transformed and expressed as log CFU/g or mL of olives or brine, respectively.

### 2.3. pH and Salt Measurement

After sampling, the pH was measured in both brine and olive samples using a digital pHmeter (model RL150, Russel Inc., Boston, MA, USA). The pH of the brine was recorded by immersing the electrode of the instrument directly in the brine sample, whereas the pH of olives was measured in a fluid slurry that was prepared by homogenizing 50 g of depitted olives in 100 mL of distilled water using an Ultra Turrax T25 blender (IKA, Staufen, Germany) [29]. Salt (sodium chloride) determinations in the brines were carried out by titration as described elsewhere [30]. The results are expressed as a percentage (*w/v*) of NaCl.

### 2.4. Total DNA Extraction

Microbial DNA from olive samples was extracted based on the protocol of Medina et al., slightly modified [22]. In detail, 12 g of olive sample were homogenized with 30 mL of sterile Ringer’s solution in a stomacher apparatus for 60 s at room temperature. After centrifugation (10,000 g/10 min/20 °C), the pellet was washed twice with 30 mL of phosphate-buffered saline (PBS) pH 7.4, and the DNA was extracted using the DNeasy PowerSoil Kit (Qiagen, Valencia, CA, USA) according to the manufacturer’s instructions. DNA was eluted in 30 μL of preheated (70 °C) DNA-free PCR-grade water and stored at −20 °C until further analysis. In addition, microbial DNA from brine samples was extracted according to a combined protocol of Pitcher et al. [31] and Kopsahelis et al. [32]. In brief, 10 mL of brine sample was centrifuged (10,000 rpm/10 min/20 °C) and the pellet was washed twice with 1 mL of PBS. After centrifugation (10,000 g/10 min/20 °C), the pellet was resuspended in 1 mL of PBS and incubated at 65 °C for 10 min to decrease the content of PCR inhibitors. The sample was centrifuged under the same conditions, the supernatant was removed, and then 500 μL of freshly prepared lysozyme (Sigma-Aldrich Chemie Gmbh, Munich, Germany) (50 mg/mL) in Tris-EDTA (TE) buffer (10 mM Tris-HCl, 1 mM EDTA, pH 8.0), 100 μL RNAse A (Sigma-Aldrich) (10 mg/mL), 40 μL mutanolysin (Sigma-Aldrich) (5 U/μL), 500 μL of 1 M sorbitol, 0.1 M EDTA (pH 7.5) and 200 μL lyticase (Sigma-Aldrich) (1 U/μL) were added to the pellet and vortexed briefly until it was completely dissolved. The suspension was incubated at 37 °C for 3 h, and, subsequently, 20 μL proteinase K (Sigma-Aldrich) (25 mg/mL) was added and an incubation at 55 °C for 1 h followed. The lysate was centrifuged at 10,000 g/10 min/20 °C, the pellet was resuspended in 0.5 mL of 50 mM Tris-Cl (pH 7.4), 20 mM EDTA and 50 μL of 20% *v/v* SDS and the mixture was incubated at 65°C for 30 min. Afterwards, 250 μL cold ammonium acetate (7.5 mol/L) was added and the tube was held on ice for 1 h. After centrifugation (10,000 g/10 min/4 °C), 1 mL of the supernatant was transferred to a new Eppendorf tube, 1 volume of cold isopropanol was added and the tube was kept overnight at -20 °C. The next day, the fibrous DNA was pelleted by centrifugation (10,000 g/20 min/4 °C), washed twice with 700 μL ice-cold ethanol (70% *v/v*) and, after the complete removal of ethanol, the pellet was resuspended in 30–50 μL of TE buffer (pH 8.0) and stored at −20 °C until use.

DNA concentration was measured using a Quawell Q5000 Read First photometer (Quawell Technology Inc, San Jose, CA, USA), and DNA quality was checked according to Papademas et al. [33].

### 2.5. Amplicon-Based Metagenomics Analysis

Amplicon sequencing (bTEFAP^®^) was performed on the Illumina MiSeq at Molecular Research DNA (MR DNA, Shallowater, Texas) in order to evaluate the bacterial and yeast/fungal microbiota. Therefore, primers 27F (5′-AGR GTT TGA TCM TGG CTC AG-3′) and 519R (5′-GTN TTA CNG CGG CKG CTG-3′) were used for the amplification of the V1-V3 hypervariable region of the bacterial 16S rRNA gene, as well as primers ITS1F (5′-CTT GGT CAT TTA GAG GAA GTA A-3′) and ITS2R (5′-GCT GCG TTC TTC ATC GAT GC-3′), to amplify the yeast/fungal internal transcribed spacer (ITS) DNA region, namely ITS1-ITS2. The PCR conditions and purification of amplicon products were performed according to Papademas et al. [33]. Operational taxonomic units (OTUs) were defined after the removal of singleton sequences, clustered at 3% divergence (97% similarity) and taxonomically assigned using the Nucleotide Basic Local Alignment Search Tool (BLASTn) against a curated National Center for Biotechnology Information (NCBI) deriving database [34]. Subsequently, clustered OTUs were used to construct the rarefaction curves in order to assess species richness and estimate the sequencing depth. Finally, alpha-diversity indices, namely observed species, Shannon and inverse Simpson were used to evaluate microbial diversity based on richness and evenness [35]. Raw sequencing data are deposited at the European Nucleotide Archive (ENA) under the study ID PRJEB37064.

### 2.6. Statistics and Exploratory Data Analysis

One-way analysis of variance (ANOVA) followed by post hoc comparisons with Tukey’s test was undertaken for the determination of differences in microbial enumerations and pH between fermented table olives from the two regions. In addition, differences between salt concentration in the brine samples from the two regions were determined using an unpaired t-test. In all cases, *p* < 0.05 was considered to be statistically significant. Data analysis was performed using the GraphPad Prism ver. 5.0 software (GraphPad Software, San Diego, CA, USA). In addition, Hierarchical cluster analysis (HCA) was undertaken for the unsupervised discrimination of the characterized microbiota (bacteria and yeasts) from the brine and olive environment based on the two geographical sampling regions (i.e., Aitoloakarnania and Messinia/Lakonia). As input matrix, the bacterial or fungal OTU table with a relative abundances higher than 1% was used, for olive or brine samples. HCA was performed based on Euclidean distance and Ward’s linkage as similarity measure and clustering algorithm, respectively. Data were transformed by autoscaling before analysis and the results were graphically illustrated in the form of heatmaps. Heatmaps are commonly used in bioinformatics as a two-dimensional visualization technique, where data are arranged in rows and columns so that similar columns are grouped together with their similarity presented by a dendrogram [36]. Furthermore, a supervised classification technique, namely Partial Least Squares Discriminant Analysis (PLS-DA), was also used in this work. PLS-DA establishes a linear regression between the X matrix of independent variables (in our case OTUs derived from NGS analysis) and matrix Y of dependent variables (classes). HCA and PLS-DA were conducted using Metaboanalyst 4.0 [37] and XLSTAT 2013.2.07 (Addinsoft, Paris, France).

## 3. Results

### 3.1. Differences in Microbial Population, pH Values and Salt Concentration According to the Region of Origin

Out of the 29 samples of fermented cv. Kalamata natural black table olives examined, 14 samples originated from the area of Aitoloakarnania in western Greece and 15 samples were from the area of Messinia/Lakonia in southern Peloponnese. Differences in LAB populations were found according to the geographical origin of the samples. Specifically, the average value of LAB counts of olives originating from Messinia/Lakonia was 3.27 ± 0.64 log CFU/g that was significantly lower (*p* < 0.05) compared to LAB counts of olives from Aitoloakarnania, which equaled to 5.98 ± 0.43 log CFU/g (Figure 2A). A similar trend was obtained for LAB enumerated in the brine samples. In this case, the average LAB counts from brine samples from Messinia/Lakonia, amounting to 3.85 ± 0.66 log CFU/mL, was significantly lower (*p* < 0.05) than those of Aitoloakarnania (6.77 ± 0.36 log CFU/mL). It should be noted though that, in four samples from Messinia/Lakonia and one sample from Aitoloakarnania, LAB could not be enumerated in the brine and/or olives (Appendix A).

On the other hand, fewer differences were found for yeasts in Kalamata natural black olives. Specifically, the average populations of yeasts for olive and brine samples originating from the area of Messinia/Lakonia were 3.52 ± 0.50 log CFU/g and 4.51 log CFU/mL, respectively, whereas lower populations were found in the samples from Aitoloakarnania, i.e., 2.92 ± 0.33 log CFU/g and 3.94 ± 0.25 log CFU/mL. No statistical differences could be established among the olive and brine samples between the two areas, with the exception of olive samples from Aitoloakarnania that presented on average lower counts compared to brine samples from Messinia/Lakonia (Figure 2B). It needs to be noted that, in two samples from Aitoloakarnania, no yeasts were detected on the olives (Appendix A). Finally, no enterobacteria were present in any of the analyzed olives and brine samples from both regions.

Furthermore, the average values of pH in both brines and olives indicated successful lactic acid fermentation processes in the two regions. Brine samples from Aitoloakarnania and Messinia/Lakonia presented average pH values of 3.92 ± 0.06 and 3.85 ± 0.07, respectively. Slightly higher pH values were measured in olive samples from both areas, i.e., 4.14 ± 0.06 (Aitoloakarnania) and 4.09 ± 0.07 (Messinia/Lakonia) (Figure 2C). The average pH values did not significantly differ between the two regions, with the exception of the average pH value of the brine samples from Messinia/Lakonia that was significantly lower (*p* < 0.05) from the average pH value of olive samples from Aitoloakarnania. The pH values in the brines and olives of the individual samples analyzed are illustrated in Appendix A. Given that the pH values in the brine of natural black olives should not exceed 4.30 according to the “Trade standard applying to table olives” of the IOC to ensure the safety of the final product, it can be concluded that no deviations were noticed.

Moreover, the average salt concentration in the brines was 7.7% ± 0.28% and 7.5% ± 0.43% for the areas of Messinia/Lakonia and Aitoloakarnania, respectively, whereas no statistical differences could be noticed between the two geographical regions (Figure 2D). The salt concentration of the individual samples of brine analyzed is shown in Appendix A. It must be mentioned that in 21 brine samples (13 from Messinia/Lakonia and eight from Aitoloakarnania), salt concentration was in the range of 7–8% or above, which is considered as an adequate salt concentration to prevent the spoilage of olives, especially in the summer period when high temperatures prevail.

### 3.2. Amplicon-Based Metagenomics Analysis

A total of 3,188,642 bacterial raw sequences were obtained from the 58 samples (29 olives and 29 brine samples). After the removal of low-quality reads, 1,783,446 sequences (an average of 29,724 ± 1530 sequences per sample) were used for metagenomic analysis. In total, 2133 bacterial OTUs were assigned among the samples, with an average of 317 ± 131 and 271 ± 58 OTUs per olive and brine sample, respectively. On the other hand, the number of yeasts/fungal raw sequences obtained from the 58 samples was lower, i.e., 2,742,360. However, the number of sequences used for metagenomic analysis after quality control was similar to that of bacterial sequences, i.e., 1,769,686 (an average of 29,495 ± 1914 sequences per sample). Among the 58 samples, 1900 yeast/fungal OTUs were identified with an average of 303 ± 125 and 348 ± 204 OTUs per olive and brine sample, respectively.

The microbial complexity was estimated on the basis of alpha-diversity indices, namely observed, Shannon and inverse Simpson. The richness estimation according to the Observed species indicated that the bacterial and yeast/fungal microbiota of olives from Messinia/Lakonia was significantly higher (*p* < 0.05), and, according to Shannon and inverse Simpson indexes, more abundant (*p* < 0.05) than that of samples from the Aitoloakarnania region (Figure 3A,C). This was also the case for yeast/fungal diversity (both richness and evenness) of brine samples from Messinia/Lakonia and Aitoloakarnania regions (Figure 3D). On the contrary, observed species, Shannon and inverse Simpson index values for the bacterial microbiota of brine samples were relatively similar for both the Messinia/Lakonia and Aitoloakarnania regions (Figure 3B).

In addition, the rarefaction curves revealed the differences of bacterial and yeasts/fungal species richness among the samples analyzed (Appendix A). In details, rarefaction curves of both 16S and ITS data of olive and brine samples attained the saturation plateau, indicating that the sequencing depth was sufficient. As also revealed by the alpha-diversity indices, the species richness of yeasts/fungal microbiota of both olive and brine samples (Appendix A), and the bacterial microbiota of olive samples (Appendix A) was higher in the samples from Messinia/Lakonia compared to the Aitoloakarnania region, while the species richness of bacteria communities of brine samples was similar between the two geographical regions (Appendix A).

The bacterial microbiota of all the olive and brine samples analyzed was covered by 24 phyla, in which Firmicutes, Proteobacteria, Bacteroidetes and Actinobacteria were the predominant. Among Firmicutes, Lactobacillaceae was the main bacterial family identified, reaching approximately an average abundance of 35.14% ± 24.37% and 45.06% ± 43.16% among the olive and brine samples, respectively, from Messinia/Lakonia, whereas average abundances were higher in olives and brines from Aitoloakarnania, i.e., 63.53% ± 26.79% and 70.71% ± 23.39%, respectively (Figure 4). This was in accordance with the classical microbiological analysis, since LAB counts of olives and brines originating from Messinia/Lakonia were significantly lower compared to these from Aitoloakarnania region (Figure 2A). Apart from Lactobacillaceae, Celerinatantimonadaceae, Leuconostocaceae and Acetobacteraceae were more abundant families in olive and brine samples from Messinia/Lakonia than Aitoloakarnania as well. Interestingly, several families, including Propionibacteriaceae, Staphylococcaceae and Chitinophagaceae, were mainly identified in olives, while Pseudomonadaceae, Cardiobacteriaceae and Acetobacteraceae were identified in brine samples (Appendix A). At the genus level, *Lactobacillus, Celerinatantimonas, Propionibacterium, Staphylococcus* and *Leuconostoc* dominated the olive samples with varying abundances depending on the geographical region (Figure 5A and Appendix A). According to the Venn diagram constructed on the basis of the average bacterial microbiota of olive samples from Messinia/Lakonia and olive samples from Aitoloakarnania, the majority of the identified genera (25 out of 44) were shared among the olives from the two regions (Figure 5C). Among the core genera, *Lactobacillus, Celerinatantimonas, Propionibacterium, Staphylococcus, Sediminibacterium, Leuconostoc, Sphingomonas* and *Enterobacter* were the most abundant. However, 10 and 9 bacterial genera were unique in olives from Messinia/Lakonia and Aitoloakarnania, respectively, all below 1.0% abundance, except of *Serratia* (1.75% ± 6.77%) in olives from Messinia/Lakonia and *Acetobacter* (1.34 ± 5.00%) in olives from Aitoloakarnania (Appendix A). Furthermore, the bacterial microbiota of brine samples from both geographical regions consisted of 44 dominant genera, including *Lactobacillus, Pseudomonas, Sphingomonas, Suttonella, Celerinatantimonas* and *Leuconostoc*, reaching approximately 80% of the bacterial sequences (Figure 5B and Appendix A). Based on the Venn diagram, only 14 genera were shared among the brine samples from Messinia/Lakonia and Aitoloakarnania, compared to the 25 core genera of olives (Figure 5D). Brine samples from the Messinia/Lakonia region had the majority of unique genera, i.e., 18; however, only three had abundance above 1.0%, namely *Bacteroides* (1.83% ± 2.81%), *Celerinatantimonas* (4.71% ± 12.22%) and *Weissella* (1.05% ± 4.08%). On the contrary, among the 12 unique bacterial genera identified in brine samples from Aitoloakarnania, five were the most abundant, i.e., *Swaminathania* (1.83% ± 6.37%), *Acetobacter* (1.24% ± 3.82%), *Salinicola* (1.13% ± 2.16%), *Empedobacter* (1.04% ± 3.89%) and *Novosphingobium* (1.03% ± 3.84%) (Appendix A).

The yeasts/fungal community of olive and brine samples from both geographical regions was characterized by a high level of Ascomycota phylum (89.58%). Compared to bacteria, yeasts/fungal microbiota was less diverse, with Pichiaceae, Aspergillaceae, Saccharomycetaceae and Debaryomycetaceae being the dominant among the 27 families identified in all samples (Figure 6 and Appendix A). The average yeasts/fungal microbiota from olives originating from Messinia/Lakonia was similar to that of Aitoloakarnania. However, a few differences were observed between the two microbial fingerprints, such as the relatively high abundance of the Cladosporiaceae (3.68% ± 6.42%) and Ascomycota (3.30% ± 7.17%) families in Messinia/Lakonia and the Phaffomycetaceae (2.97% ± 7.10%) family in Aitoloakarnania microbiota (Figure 6A and Appendix A). Interestingly, the yeasts/fungal community of brine samples was less diverse than that of olives, including 12 families (Figure 6B and Appendix A). The average abundance of Pichiaceae family was relatively higher in brine samples from Messinia/Lakonia (74.77 ± 25.10%) compared to Aitoloakarnania (42.27% ± 36.32%), while, on the contrary, Saccharomycetaceae (25.27% ± 34.48% vs. 14.60% ± 23.87%), Debaryomycetaceae (15.73% ± 32.51% vs. 5.12% ± 10.05%), Phaffomycetaceae (8.99% ± 20.85% vs. 0.11% ± 0.41%) and Cystofilobasidiaceae (5.05 ± 15.41 vs. 0.70 ± 1.91%) families were mainly identified in the brine samples from Aitoloakarnania region. At the genus level, *Pichia, Penicillium, Ogataea, Saccharomyces, Malassezia* and *Millerozyma* were predominant in olive samples, with varying abundances depending on the geographical region (Figure 7A and Appendix A). Among the 39 yeasts/fungal genera identified in olive samples, approximately 43.6% were shared among the samples, compared to 56.81% of the bacterial core genera. Interestingly, from the total of 39 genera identified, 15 and seven were unique for olives originating from Messinia/Lakonia and Aitoloakarnania, respectively (Figure 7C and Appendix A). It should be noted though, that the abundances of the majority of the unique genera were below 1.0%, except of *Engyodontium* (3.30% ± 7.17%) and *Brettanomyces* (1.93% ± 5.54%) for Messinia/Lakonia and *Wickerhamomyces* (2.96% ± 7.10%) for Aitoloakarnania regions. As already mentioned above, the richness of the yeasts/fungal microbiota of brine samples was lower compared to that of olive samples, which is also shown in the Venn diagram (Figure 7B,D). The majority of the yeast/fungal genera identified (62.5%) were shared among the brine samples from the two regions, with *Pichia, Ogataea, Saccharomyces* and *Millerozyma* being the dominant ones (Appendix A). However, five and four genera were unique for brines from Messinia/Lakonia and Aitoloakarnania, respectively, all below 1.0% abundance, except of *Zygosaccharomyces* (7.72% ± 20.70%) for Messinia/Lakonia and *Kluyveromyces* (1.49% ± 3.96%) for Aitoloakarnania regions (Appendix A).

### 3.3. Differences in Microbial Community Structure By Multivariate Analysis

A two-dimensional graphical representation (heatmap) was employed for data illustration, in which each microbial species is characterized by a single row and each column represents an individual olive or brine sample analyzed from the two geographical regions. Specifically, Figure 8 shows the heatmap of the bacterial community for Kalamata natural black olives clustered by Euclidean distance. The clusters indicated higher bacterial diversity in both olive and brine samples originating from Messinia/Lakonia compared to the Aitoloakarnania region. Regarding the bacterial community structure of brines, five samples, namely B_1_, B_2_, B_4_, B_27_, and B_28_, from the Messinia/Lakonia area presented higher abundances in *Pseudomonas, Microbacterium, Caulobacter, Sphingomonas, Sediminibacterium, Burkholderia, Comamonas,* and *Bacteroides* species (Figure 8A). In addition, *Lactobacillus* spp. and *Leuconostoc* spp. were abundant in several brine samples from both geographical areas, whereas, on the other hand, *Lactococcus* spp. was abundant in three samples from Aitoloakarnania, namely B_18_, B_22_ and B_23_.

The bacterial community of olives from the Kalamata variety from both geographical areas is presented in Figure 8B, with *Lactobacillus* spp. and *Leuconostoc* spp. being abundant in several samples from both areas. Two samples from Aitoloakarnania, namely O_2_ and O_16_, presented a high abundance in enterobacteria (*Pantoea* spp., *Serratia* spp., *Klebsiella* spp. and *Enterobacter* spp.), whereas *Corynebacterium* spp., *Propionibacterium* spp. and *Staphylococcus* spp. were mostly abundant in olive samples from Messinia/Lakonia. It needs to be noted that *Staphylococcus* spp. was highly correlated with four samples (O_1_, O_4_, O_6_ and O_12_) from Messinia/Lakonia, a fact that should raise concern about the hygiene conditions in these fermentations.

The structure of the yeasts/fungal community of olive and brine samples from both regions is illustrated in Figure 9. Higher fungal diversity can be observed in fermented olives compared to brines regardless of the geographical region of origin. The most represented genus in the brines of samples from Messinia/Lakonia was *Pichia* followed by *Candida, Zygosaccharomyces, Zygoascus, Zygotorulaspora, Brettanomyces,* and *Schanniomyces.* On the contrary, brine samples from Aitoloakarnania were abundant in *Saccharomyces* spp., *Wickerhamomyces* spp., and *Ogataea* spp. (Figure 9A). A less clear clustering pattern between the two regions could be obtained for the yeasts/fungal community of fermented olives. The most abundant yeasts/fungi from the area of Messinia/Lakonia were *Pichia* spp., *Candida* spp., *Aureobacidium* spp., *Alternaria* spp., *Penicillium* spp., *Cladosporium* spp., and *Phyllactinia* spp., while *Saccharomyces* spp. and *Ogataea* spp. were highly related to fermented olives from Aitoloakarnania (Figure 9B).

PLS-DA analysis using species-level OTUs revealed several differences in microbial communities among the olive and brine samples analyzed. The PLS-DA scores plot for the bacterial microbiota in the brines showed discrimination between the two geographical regions, although an overlapping was evident among a number of samples (Figure 10A). In addition, the loadings plot showed the correlation of each bacterial species to the brine samples from the two regions (Figure 10B). It can be concluded that brine samples from Aitoloakarnania (A) were highly correlated with *Lactobacillus* spp., whereas brine samples from Messinia/Lakonia (M) were associated with *Pseudomonas* spp., *Microbacterium* spp., *Caulobacter* spp., *Burkholderia* spp. and *Sphingomonas* spp. Furthermore, the influence of bacterial species (X-variable) on the geographical regions (Y-response) can be also expressed with the Variable Importance in Projection (VIP) coefficient which, in our case, indicates the bacterial species having the highest importance in explaining the Y-variance (i.e., the origin of the samples). A VIP value of 1.0 has generally been accepted as a cut-off limit in variable selection; thus, variables exceeding this limit can be considered to be highly influential. Based on the VIP values, *Microbacterium* spp., *Sphingomonas* spp., *Burkholderia* spp., *Bacterioides* spp., *Pseudomonas* spp., *Caulobacter* spp., *Pelomonas* spp., *Sediminibacter* spp., and *Celerinatantimonas* spp. were highly associated with the brine samples from the area of Messinia/Lakonia (Figure 11). On the contrary, *Lactobacillus* spp., *Salinicola* spp., *Lactococcus* spp., *Idiomarina* spp., and *Martellela* spp. were highly correlated with the brine samples from the area of Aitoloakarnania. Furthermore, the discrimination profile according to the PLS-DA analysis for the bacterial communities in the olive samples was similar to that of brines; however, it should be noted that the overlapping observed in brine samples was absent in olives (Appendix A). Interestingly, a different pattern of influential bacterial species was observed on the olive samples compared to the brines, based on the VIP scores. Specifically, olive samples from Messinia/Lakonia were highly associated with *Staphylococcus* spp., *Propionibacterium* spp., *Micrococcus* spp., *Corynebacterium* spp., *Celerinatantimonas* spp., *Arcicella* spp., *Streptococcus* spp., *Rubrobacter* spp., and *Prevotella* spp., while, on the contrary, olive samples from Aitoloakarnania were highly correlated with *Lactobacillus* spp., *Lactococcus* spp., and *Leuconostoc* spp. (Appendix A).

On the other hand, the PLS-DA analysis based on the yeasts/fungal community structure provided a clear discrimination among the brine samples from the two geographical areas (Figure 12A). The absence of overlapping among the brine samples indicates that the yeasts/fungal community could be used more effectively in the discrimination of the sample’s origin compared to the bacteria community. In addition, the plot of loadings revealed that the brine samples from the area of Messinia/Lakonia were highly associated with *Pichia* spp., *Candida* spp., *Cryptococcus* spp., *Cystofilobasidium* spp., and *Brettanomyces* spp., whereas *Wickerhamomyces* spp., *Debaryomyces* spp., *Kluyveromyces* spp. and *Saccharomyces* spp. were correlated with the brine samples from the area of Aitoloakarnania (Figure 12B). The highly influential yeasts/fungal species were further identified by the VIP scores (cut off value of 1) [38], in which *Pichia* spp. was also found to be highly correlated with brine samples from Messinia/Lakonia, together with *Brettanomyces* spp., *Candida* spp., *Zygosaccharomuces* spp., and *Zygotorulaspora* spp. On the contrary, for brine samples from Aitoloakarnania, high correlations were observed for *Wickerhamomyces* spp. and *Kluyveromyces* spp. (Figure 13).

Moreover, the score plot of the PLS-DA analysis for the yeasts/fungal communities, although it discriminated the olive samples from the two geographical regions, revealed an overlap among some of them (Appendix A). According to the VIP scores, *Pichia* spp., *Candida* spp., and *Brettanomyces* spp. were also highly correlated with the olive samples from Messinia/Lakonia, as in the case of brine samples from the same area *Cladosporium* spp., *Alternaria* spp., and *Engyodontium* spp. Similar to the brines, *Wickerhamomyces* spp. and *Kluyveromyces* spp. were found to be highly correlated with the olives from Aitoloakarnania, along with *Ogatae* spp. (Appendix A).

## 4. Discussion

In this study, the bacterial and yeast/fungal microbiota of the olives and brine samples of cv. Kalamata natural black olives were elucidated using amplicon-based metagenomic analysis. Samples were industrially fermented according to the traditional Greek-style method and obtained from the two main producing regions of this cultivar, namely Messinia/Lakonia in southern Peloponnese and Aitoloakarnania in western Greece. Due to the highly diverse terrain and climate conditions between the two regions, the potential microbial biogeography association between certain taxa and geographical area was also assessed. A critical factor that affects the fermentation process and determines the survival of LAB during the fermentation is the salt concentration in the brine. It has been reported that high salt levels (>8%) could favor the dominance of yeasts and inhibit the growth of LAB [39], rendering a final product with less preservation characteristics. Nowadays, the Greek table olive industry has reduced the salt level in the brine to 6–7% during the period of active fermentation to ensure the dominance of LAB and thus improve the preservation and sensory attributes of the final product. According to our analysis, the average salt concentration in the brine samples from both regions was less than 8% (Figure 2D), although great variations were observed in the individual samples (Appendix A), indicating that further measures must be undertaken by the industry to standardize the fermentation process so as to improve the quality of the final product.

As already mentioned, the co-existence of bacterial and yeast/fungal communities is of fundamental importance during the fermentation process. These communities could be considered as a natural resource associated with a specific agroecosystem, shaping the microbial “terroir” of table olives [40]. The term “terroir”, although initially employed in oenology, has been extended, nowadays, to other crops and food products to link geographical origin and environmental conditions to quality aspects of agricultural commodities [41]. According to the alpha-diversity analysis, the diversity (both richness and evenness) of the yeast/fungal microbiota of both olive and brine samples (Figure 3C,D), and the bacterial microbiota of olives (Figure 3A), was higher in the samples from Messinia/Lakonia compared to the Aitoloakarnania region, while bacteria communities of brine samples (Figure 3B) were similar between the two geographical regions. Interestingly, based on the identified OTUs, the yeasts/fungal microbiota of both olives and brines was less diverse compared to bacteria. This could be explained since the olives were spontaneously fermented by the indigenous microbiota, which is subjected to seasonal variations [42], as well as to a variety of contaminating microorganisms from fermentation vessels, pipelines, pumps and other equipment in contact with olives and brine, that could be considered as a persistent microbiota in the factory facilities [18,22,43].

According to the results of the 16S metagenomics analysis, Lactobacillaceae was the dominant family identified in olive and brine samples from both regions (Figure 4). The high average abundance of the family Lactobacillaceae in the olive samples from both regions confirmed the ability of LAB to colonize the surface of olives [44,45]. Among the family Lactobacillaceae, *Lactobacillus* and *Leuconostoc* were the most abundant genera identified, indicating that these were all lactic acid fermentations, as also verified by the classical microbiological analysis (Figure 2). Both genera are commonly found in the microbiota of fermented green and black olives using both classical microbiological and metagenomics analyses [6,17,20,22,23,24]. This observation updates the findings of other researchers who reported that LAB could not be detected throughout the spontaneous fermentation of Kalamata variety or at least they could be detected only in the initial stage of the process [46,47]. In a recent work, the bacterial and yeast/fungal microbiota of brines from three Greek olive varieties, namely Kalamata, Conservolea and Kerasoelia, originating from different geographical regions of Greece, was assessed using an amplicon-based metagenomic technique [26]. The authors reported the dominance of the genus *Lactobacillus* (74%) in the Kerasoelia variety, whereas in the Kalamata and Conservolea varieties, the genus *Cellulosimicrobium* prevailed, with abundances ranging from 39% to 90%. In our samples, the genus *Cellulosimicrobium* was found in only one brine sample, namely B_27_, from the region of Messinia/Lakonia with relatively low abundance, i.e., 1.26% (Appendix A). The almost absence of the genus *Cellulosimicrobium* in our samples could be attributed to the different geographical regions, as the olive samples analyzed in the work of Papadimitriou et al. [26] originated from the island of Evoia in the eastern part of mainland Greece, thus strengthening the hypothesis that the conditions of different agroecosystems could have a decisive role in the diversity of microbial communities [48].

Moreover, it is noteworthy that the genus *Staphylococcus* was identified in four olive samples from the area of Messinia/Lakonia, with abundances ranging from 8% to 17% (Appendix A). As some species of this bacterial genus have been characterized as opportunistic pathogens [49], special attention should be given to the hygiene conditions in these specific fermentation vessels to avoid the adaptation of this species to the processing environment of olives. Consequently, intense cleaning would be advisable in these fermenters to avoid persistence of undesirable bacteria, although some researchers report that intense hygiene conditions may also negatively affect the establishment of a favorable autochthonous microbiota in the processing environment [18]. However, our results concerning the identification of opportunistic pathogens should be evaluated down to species or even strain level. The presence of the genus *Staphylococcus* in low abundances has been also confirmed by other researchers in directly brined green cracked *Aloreña de Málaga* olive fermentations [22], as well as in Spanish-style green olive fermentation of Manzanilla and Gordal varieties on an industrial scale [18,40,50]. No other food-related bacterial pathogens, such as *Listeria, Clostridium, Salmonella* and *Escherichia* were detected in any of the olive and brine samples analyzed from both regions. This finding is of paramount importance regarding the safety of table olives as the outbreaks due to table olive consumption are scarce. Our results are in accordance with other studies that employed metagenomics approaches to elucidate the bacterial community structure in Italian and Spanish olive varieties and reported the absence or low abundance of bacterial potential pathogens [17,20,22].

Furthermore, the family Enterobacteriaceae was found in several olives and brines from both regions, with *Enterobacter* being the most abundant genus. However, the majority of olive and brine samples analyzed presented relatively low abundances of this family, with the exception of two olive samples, namely O_2_ and O_16_, from Messinia/Lakonia and Aitoloakarnania, respectively. The presence of this family in the fermentation of table olives is rather habitual, with a well-known negative contribution in the quality of the final product [42]. However, it should be noted that, according to the classical microbiological analysis, no enterobacteria were counted in any of the analyzed samples. The absence of culturable enterobacteria can be attributed to the low pH values of brines and olives that ensure the safety of the final product. Thus, the identification of the family Enterobacteriaceae in several samples highlights the disadvantage of DNA-based metagenomics techniques, which is the amplification of DNA that may arise from dead or compromised cells [12]. Luckily, Proteobacteria, including Enterobacteriaceae, are inhibited in pH values below 4.5, and thus olives are protected from spoilage (gas pockets) due to the outgrowth of this microbial group during the fermentation [27,51]. Among the Gram-negative bacteria, the genera *Pseudomonas* and *Sphingomonas* were detected in all brine samples from both regions. The presence of the genus *Pseudomonas* has been initially detected on the surface of olives and the brines during black olive fermentation using culture dependent methods [39,52,53]. Recently, the genus *Pseudomonas* was also found in directly brined green olives of the *Aloreña de Málaga* variety using culture-independent techniques [14,22]. The presence of this proteolytic genus in table olives could have a negative effect on the overall stability and safety of the final product, either directly through the reduction of acidity in the brines and swelling of cans [54], or indirectly through the development of biogenic amines [14]. Moreover, the presence of the genus *Sphingomonas* has been reported for the first time in directly brined *Alorena* green olives fermented on an industrial scale in southern Spain [14], as well as in the Greek-type fermentation of *Bella di Cerignola* green olives [20]. The genus *Sphingomonas* is strictly aerobic, widely distributed in soil and water, as well as in plant root systems due to its ability to survive in low concentration of nutrients [55]. In addition, the genus *Propionobacterium* was also found in our olive samples, with varying abundances depending on the geographical region. This genus is associated with undesirable secondary fermentations, where the lactic acid produced during the main fermentation process is assimilated by propionic acid bacteria producing propionic acid, acetic acid and CO_2_, resulting in a gradual increase in pH and volatile acidity values, with the concurrent development of off-odors and undesirable sensory perceptions in table olives (zapateria) [27]. The control of salt concentration and pH values is thus necessary to eliminate the activity of these microorganisms, especially during the summer months [30].

The yeast/fungal microbiota of olive and brine samples from both geographical regions was less diverse compared to bacteria. Among the filamentous fungi identified, the genus *Penicillium* presented the highest abundances in the olive and brine samples from both regions (Appendix A). This result is in agreement with a recent work [56] that investigated the fungal community associated with fermented black olives of Italian and Greek olive varieties, including Kalamata natural black olives. The authors isolated and identified 60 strains as *Penicillium crustosum, Penicillium roqueforti, Penicillium paneum, Penicillium expansum, Penicillium polonicum* and *Penicillium commune.* It has been documented that the presence of filamentous fungi on olives is associated with various mycotoxins (ochratoxin, aflatoxin B, and citrinin) in green and black olives [57,58,59], but their concentration is too low to pose a risk to human health [60]. With regard to the yeast community, *Pichia*, *Saccharomyces, Ogatea* and *Millerozyma* were the dominant genera identified in both brine and olive samples (Appendix A). The presence of the former two yeast genera is well-documented in the fermentation of natural black olives of Kalamata and Conservolea varieties [26,46,61,62]. The role of yeasts in table olive production could enhance or deteriorate the quality of the final product. On the positive side, they are present throughout the fermentation process and produce compounds with important sensory attributes determining the quality of fermented olives. On the negative side, they can also be spoilage microorganisms during olive fermentation and subsequent storage and packing causing gas pockets, swollen containers, cloudy brines and off-flavors and odors [63]. The presence of the genus *Pichia* in the fermentation may have a positive impact, as several strains have antioxidant activity that protects olives from oxidation and peroxide formation [64]. In addition, several strains belonging to this genus present strain-specific killer activity against spoilage yeasts and could thus influence fermentation by affecting the yeast association during the process [65,66]. Regarding the genus *Saccharomyces*, it has been reported that some strains may have lipase activity and increase the free fatty acid content in olives during fermentation [67], whereas other strains may synthesize bioactive compounds, such as carotenoids, tocopherols, citric acid and glutathione, with interesting antioxidant potential [68].

Additionally, we used the bacterial and yeast/fungal microbiota of olives and brines as “microbial fingerprints” to discriminate the samples according to their geographical origin. The outcome of the PLS-DA analysis illustrated that a satisfactory discrimination could be obtained between the two regions, although some degree of overlapping could not be avoided. The best discrimination was obtained by the ITS data of the brine samples (Figure 12). Based on the VIP values derived from PLS-DA analysis, the most discriminative genus among the yeasts/fungal communities was *Pichia* for both brine and olive samples that were highly associated (VIP > 2.5 and 2.0 for brine and olive samples, respectively) with the region of Messinia/Lakonia (Figure 12 and Appendix A). Regarding bacterial communities, the genus *Lactobacillus* was highly influential (VIP > 2.2) for the differentiation of olive samples from the area of Aitoloakarnania (Appendix A). These observations highlight the influence of the microclimate conditions on the microbiota of, not only fermented olives [40], but also of other food commodities as well [69,70].

## 5. Conclusions

The results obtained from the present study reveal the complex microbial community structure of bacterial and yeast/fungal microbiota of 29 olives and brines of cv. Kalamata olives industrially fermented in the Messinia/Lakonia and Aitoloakarnania regions. To the best of our knowledge, this is the first systematic study on the microbial biogeography of Kalamata variety natural black olives from the two main producing regions of Greece, using the state-of-the-art approach of metagenomics. However, further studies on the occurrence of these taxa should be undertaken in other table olive industries in the same geographical regions to enhance our knowledge on the microbial communities present during Kalamata olives fermentation and their role on the quality and safety of the final product. Finally, our better understanding on the microbial composition of Kalamata table olives would result in the development of specific starter cultures to promote unique organoleptic characteristics in the final product of each region.

## Figures and Tables

**Figure 1 microorganisms-08-00672-f001:**
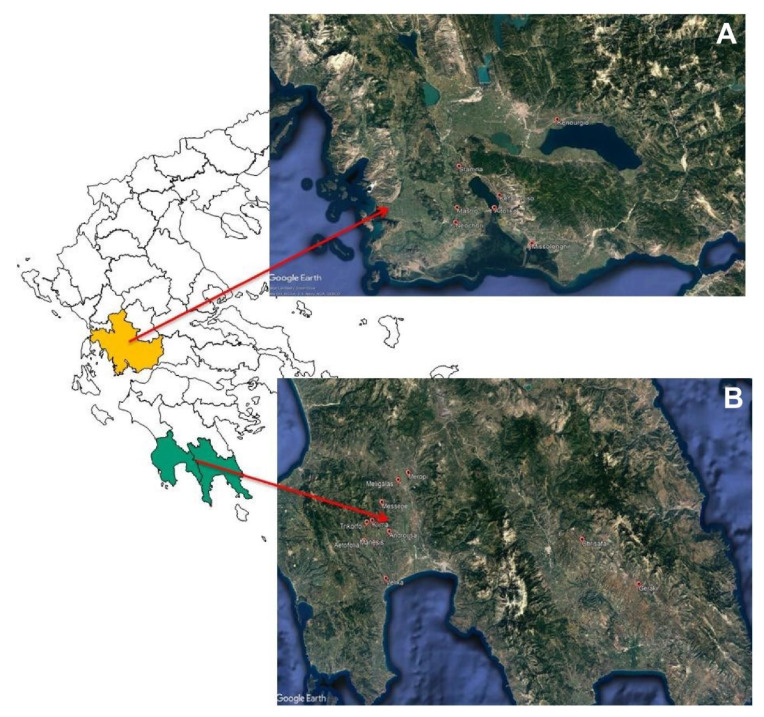
Geographical origin of fermented cv. Kalamata natural black olives from (**A**) Aitoloakarnania (western Greece) and (**B**) Messinia/Lakonia (southern Peloponnese).

**Figure 2 microorganisms-08-00672-f002:**
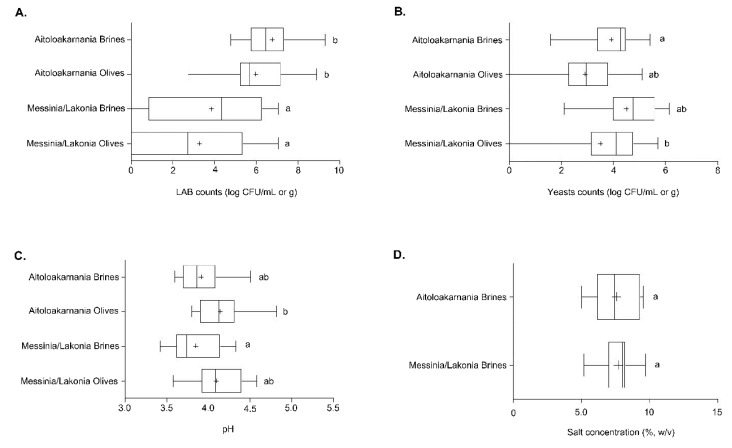
Boxplots of lactic acid bacteria (LAB) (**A**) and yeast (**B**) counts as well as pH values (**C**) from the olives and brines, along with salt concentration (**D**) from the brines of natural black cv. Kalamata olives from the two geographical regions. Different letters indicate statistically significant differences (*p* < 0.05).

**Figure 3 microorganisms-08-00672-f003:**
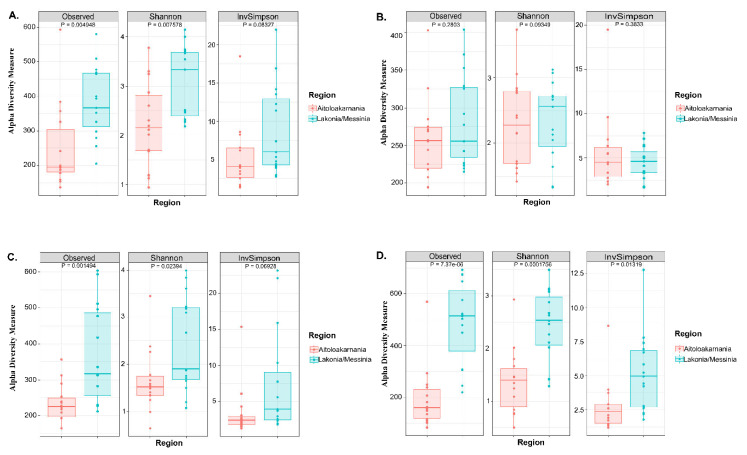
Boxplots of alpha-diversity indices, namely observed, Shannon and inverse Simpson for bacterial communities in olive (**A**) and brine (**B**) samples, as well as for yeast/fungal microbiota in olives (**C**) and brines (**D**). Samples are colored according to the two geographical regions, i.e., pink for Aitoloakarnania and light blue for Messinia/Lakonia.

**Figure 4 microorganisms-08-00672-f004:**
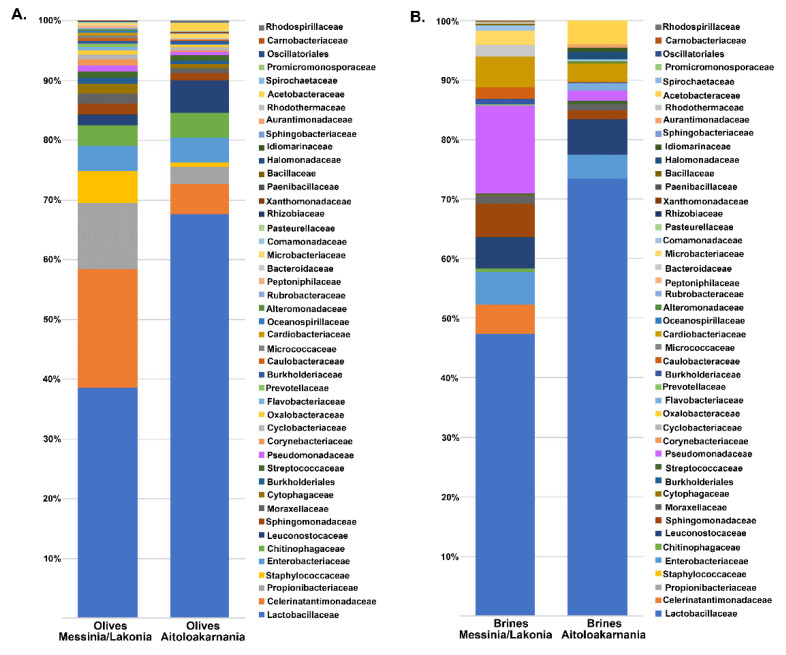
Average relative abundance (%) of dominant bacterial families obtained by 16S metagenomics analysis of olive (**A**) and brine (**B**) samples from the Messinia/Lakonia and Aitoloakarnania regions.

**Figure 5 microorganisms-08-00672-f005:**
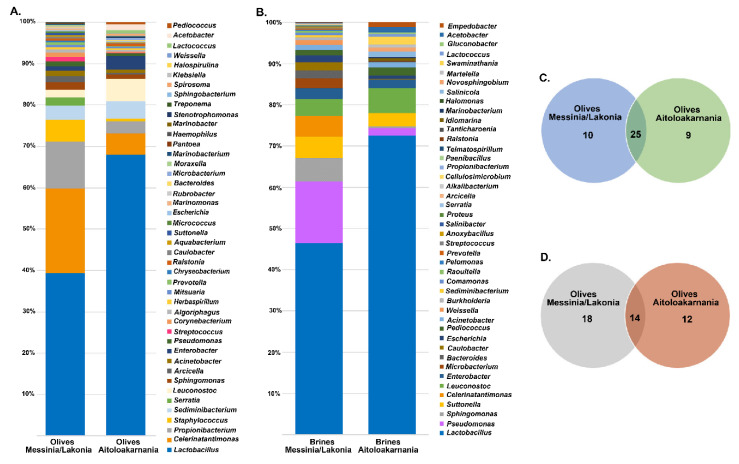
Average relative abundance (%) of dominant bacterial genera obtained by 16S metagenomics analysis of olive (**A**) and brine (**B**) samples from the Messinia/Lakonia and Aitoloakarnania regions. Venn diagram showing the number of unique and shared bacterial genera among olives (**C**) and brines (**D**) from the two geographical regions.

**Figure 6 microorganisms-08-00672-f006:**
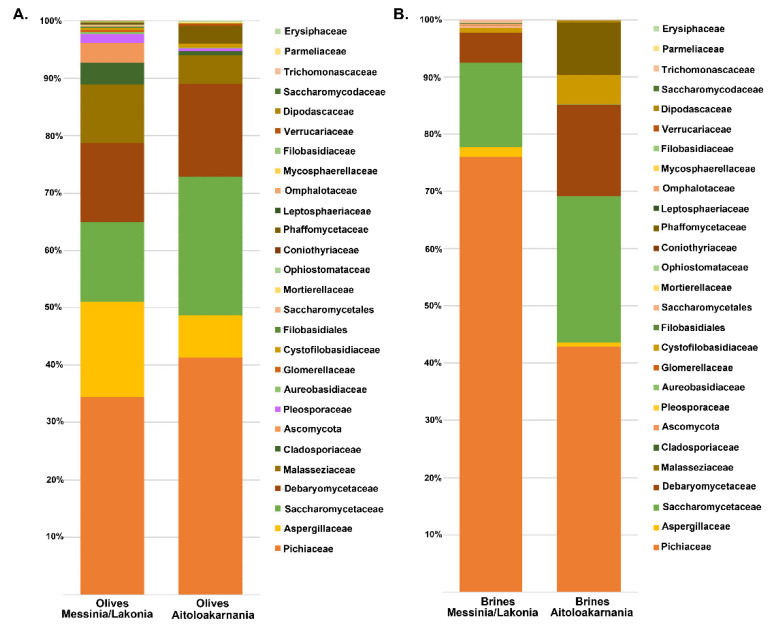
Average relative abundance (%) of dominant yeast/fungal families obtained by internal transcribed spacer (ITS) metagenomics analysis of olive (**A**) and brine (**B**) samples from the Messinia/Lakonia and Aitoloakarnania regions.

**Figure 7 microorganisms-08-00672-f007:**
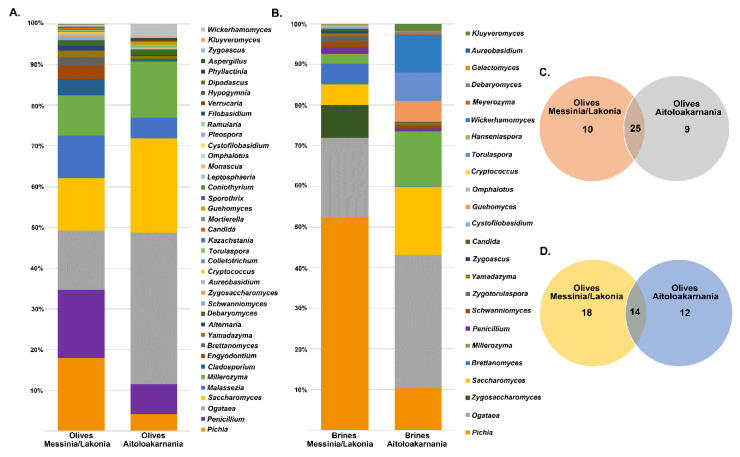
Average relative abundance (%) of dominant yeast/fungal genera obtained by ITS metagenomics analysis of olive (**A**) and brine (**B**) samples from Messinia/Lakonia and Aitoloakarnania regions. Venn diagram showing the number of unique and shared yeast/fungal genera among olives (**C**) and brines (**D**) from the two geographical regions.

**Figure 8 microorganisms-08-00672-f008:**
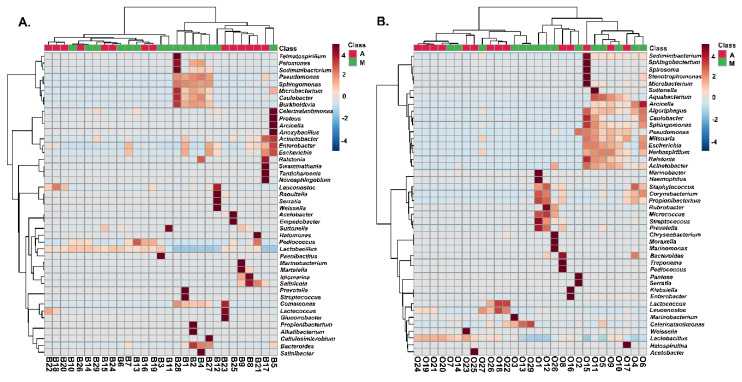
Hierarchically clustered heatmap of the bacterial community from cv. Kalamata natural black olives from the brine (**A**) and olives (**B**). Classes A and M correspond to the samples from the region of Aitoloakarnania (A) and Messinia/Lakonia (M), respectively. The sample codes are indicated in Table 1.

**Figure 9 microorganisms-08-00672-f009:**
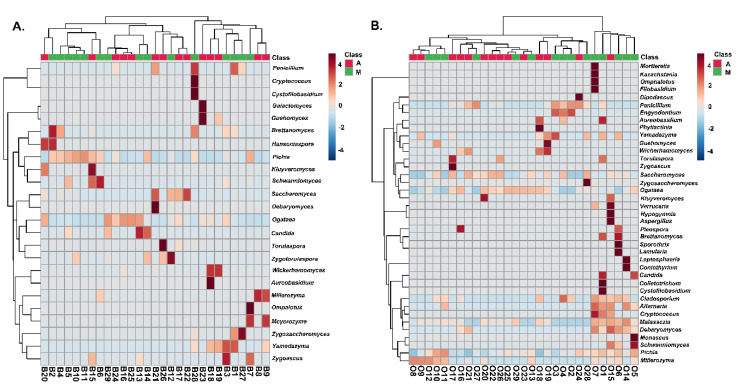
Hierarchically clustered heatmap of the yeasts/fungal community from cv. Kalamata natural black olives from the brine (**A**) and olives (**B**). Class A and M corresponds to the samples from the region of Aitoloakarnania (A) and Messinia/Lakonia (M), respectively. The sample codes are indicated in Table 1.

**Figure 10 microorganisms-08-00672-f010:**
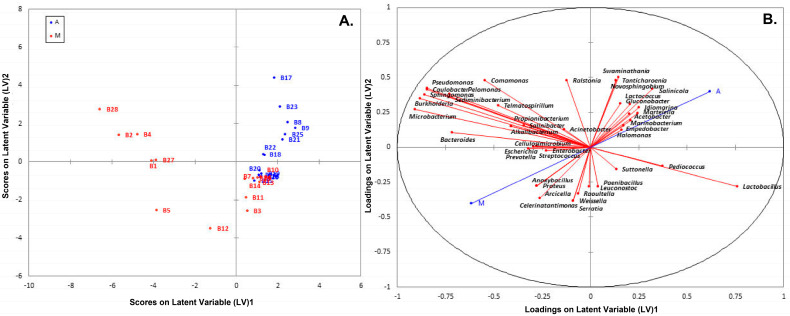
Plot of scores (**A**) and loadings (**B**) of the first two latent variables of the PLS-DA model of the bacterial community build on the brine samples from the region of Aitoloakarnania (A) and Messinia/Lakonia (M).

**Figure 11 microorganisms-08-00672-f011:**
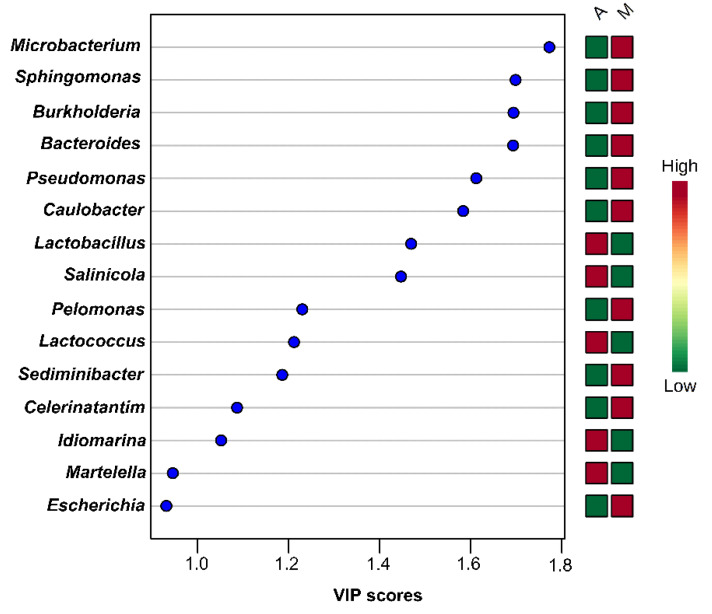
Most influential bacterial genera of the brine samples from the region of Aitoloakarnania (A) and Messinia/Lakonia (M) based on the Variable Importance in Projection (VIP) scores from the Partial Least Squares Discriminant Analysis (PLS-DA) analysis. The color bars indicate the intensity of thevariable in the respective group.

**Figure 12 microorganisms-08-00672-f012:**
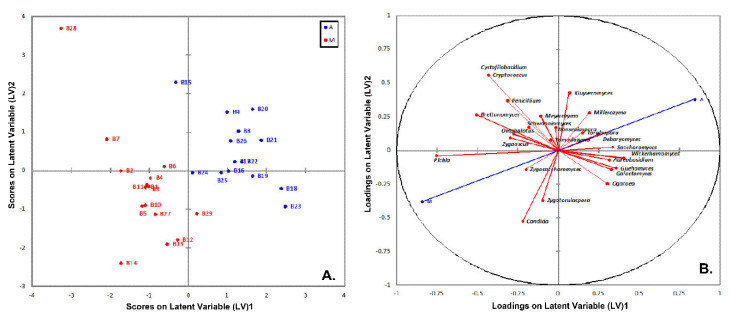
Plot of scores (**A**) and loadings (**B**) of the first two latent variables of the PLS-DA model of the yeasts/fungal community build on the brine samples from the region of Aitoloakarnania (A) and Messinia/Lakonia (M).

**Figure 13 microorganisms-08-00672-f013:**
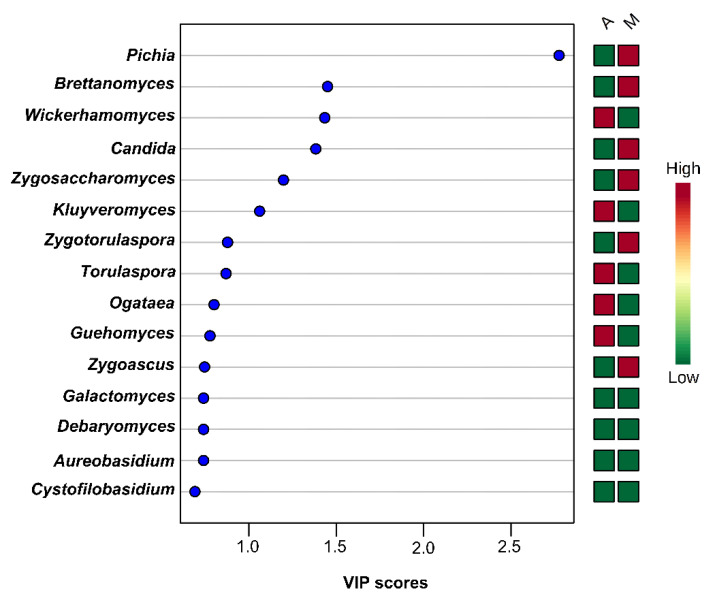
Most influential yeasts/fungal genera of the brine samples from the region of Aitoloakarnania (A) and Messinia/Lakonia (M) based on the VIP scores from the PLS-DA analysis. The color bars indicate the intensity of the variable in the respective group.

**Table 1 microorganisms-08-00672-t001:** Sample code and geographical origin of the fermented cv. Kalamata natural black olive samples.

Sample Code	Sample Origin	Sample Code	Sample Origin
1	Aetofolia/Messinia	16	Messolongi/Aitoloakarnania
2	Geraki/Lakonia	17	Aitoliko/Aitoloakarnania
3	Manesis/Messinia	18	Stamna/Aitoloakarnania
4	Velika/Messinia	19	Kefalovriso/Aitoloakarnania
5	Geraki/Lakonia	20	Chrysovergi/Aitoloakarnania
6	Messene/Messinia	21	Kainourgio/Aitoloakarnania
7	Trikorfo/Messinia	22	Aitoliko/Aitoloakarnania
8	Aitoliko/Aitoloakarnania	23	Mastro/Aitoloakarnania
9	Neochori/Aitoloakarnania	24	Neochori/Aitoloakarnania
10	Androusa/Messinia	25	Neochori/Aitoloakarnania
11	Klima/Messinia	26	Stamna/Aitoloakarnania
12	Meligalas/Messinia	27	Chrisafa/Lakonia
13	Meropi/Messinia	28	Geraki/Lakonia
14	Klada/Messinia	29	Trikorfo/Messinia
15	Messolongi/Aitoloakarnania

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
