# Peer review of "Unraveling the Microbiota of Natural Black cv. Kalamata Fermented Olives through 16S and ITS Metataxonomic Analysis"

_microorganisms, 2020, doi:10.3390/microorganisms8050672_

Round 1

Reviewer 1 Report

I have just few minor comments:

The quality of Figures 6- 14 and 16 should be improved.

Figure 7 (rarefaction curves) should be considered as a supplementary figure.

Figures 15 and 17: please check the genera names which are truncated in some cases and correct where necessary; moreover, please write all genera names in italic font and with initial capital letter.

Line 72: delete were

Line 115: correct “enterobacreria” to “enterobacteria”

Line 121: delete “401728”

Line 150: correct “centrifuge” to “centrifuged”

Line 280: correct “significant” to “significantly”

Line 288 (Figure 6 legend): correct “bacteria” to “bacterial”

Line 324: change “nine” with “9”

Line 527: change “confirming” to “confirmed”

Lines 528-531: this sentence is not clear. The Authors should rephrase it trying to better highlight the relation between the most abundant LAB genera identified and lactic fermentation

Lines 565-572: the Authors should add a comment concerning the discrepancy between the results obtained by microbiological assays, which did not reveal the presence of Enterobacteriaceae (see lines 234-235), and those obtained through metagenomic analysis

Line 632: correct “of” with “from”

Author Response

We would like to thank the Reviewer for his/her comments and suggestions that helped us to improve our manuscript. Please find our responses below.

Comment 1: The quality of Figures 6- 14 and 16 should be improved.

Reply: The quality of all Figures has been improved.

Comment 2: Figure 7 (rarefaction curves) should be considered as a supplementary figure.

Reply: Figure 7 is now presented as Supplementary Figure 6 according to the Reviewer’s suggestion

Comment 3: Figures 15 and 17: please check the genera names which are truncated in some cases and correct where necessary; moreover, please write all genera names in italic font and with initial capital letter.

Reply: The names in Figures 15 and 17 (in the revised manuscript Figures 11 and 13) have been corrected

Comment 4: Line 72: delete were

Reply: “were” was deleted

Comment 5: Line 115: correct “enterobacreria” to “enterobacteria”

Reply: “enterobacreria” was corrected

Comment 6: Line 121: delete “401728”

Reply: “401728” was deleted

Comment 7: Line 150: correct “centrifuge” to “centrifuged”

Reply: “centrifuge” was corrected

Comment 8: Line 280: correct “significant” to “significantly”

Reply: “significant” was corrected

Comment 9: Line 288 (Figure 6 legend): correct “bacteria” to “bacterial”

Reply: “bacteria” was corrected

Comment 10: Line 324: change “nine” with “9”

Reply: “nine” was corrected

Comment 11: Line 527: change “confirming” to “confirmed”

Reply: “confirming” was corrected

Comment 12: Lines 528-531: this sentence is not clear. The Authors should rephrase it trying to better highlight the relation between the most abundant LAB genera identified and lactic fermentation

Reply: Lines 524-529 in the revised manuscript have been accordingly revised as follows:

“The high average abundance of the family Lactobacillaceae in the olive samples from both regions confirmed the ability of LAB to colonize the surface of olives [44,45]. Among the family Lactobacillaceae, Lactobacillus and Leuconostoc were the most abundant genera identified, indicating that these were all lactic acid fermentations, as also verified by the classical microbiological analysis (Figure 2).”

Comment 13: Lines 565-572: the Authors should add a comment concerning the discrepancy between the results obtained by microbiological assays, which did not reveal the presence of Enterobacteriaceae (see lines 234-235), and those obtained through metagenomic analysis

Reply: We have now added the following comment according to the Reviewer’s suggestion (please see lines 568-573 in the revised manuscript):

“However, it should be noted that according to the classical microbiological analysis no enterobacteria were counted in any of the analyzed samples. The absence of culturable enterobacteria can be attributed to the low pH values of brines and olives that ensure the safety of the final product. Thus, the identification of the family Enterobacteriaceae in several samples highlights the disadvantage of DNA-based metagenomics techniques, which is the amplification of DNA that may arise from dead or compromised cells [12].”

Comment 14: Line 632: correct “of” with “from”

Reply: “of” was corrected

Reviewer 2 Report

Ms. ID Microorganisms-792130: Unraveling the microbiota of natural black cv.

Kalamata fermented olives through 16S and ITS metagenomic analysis

General comments:

This study helps to better understand the microbiology of fermented olives commonly found in Greece using classical microbiological methods and a metataxonomic approach to know the bacterial and fungal communities.

The manuscript is generally well written and the metataxonomic analyses were well interpreted. For the reader less familiar with this fermented product, this study fills in some of the gaps.

However, the manuscript contains a large number of figures, some of which could be grouped together or added to the supplemental material.

Minor Comments: 

Title: Please replace metagenomics by metataxonomic.

Comments: Metataxonomics is a term proposed by Marchesi and Ravel and is defined as the high-throughput process used to characterize the entire microbiota and create a metataxonomic tree, which shows the relationships between all sequences obtained. “Metataxonomic analysis, because it relies on the amplification and sequencing of taxonomic marker genes, is not metagenomics”. Metagenomics refers to shotgun sequencing of DNA extracted from a sample

MARCHESI, Julian R. et RAVEL, Jacques. The vocabulary of microbiome research: a proposal. 2015.

Material and Methods

Lines 176-177: Please specify the sequencing platform: Illumina MiSeq, Ion S5 XL or PAC BIO SEQUEL?  

Results:

Figures 2, 3, 4 and 5 should be grouped in Figure 2a,b,c, and d.  

Figures 6, 8, 9, 10, 11, 12, and … the quality should be improved

Figure 7 should be added in the supplemental material.

Figures 15 and 17 X-axis, The name of microorganisms should be modifier. Capital letter, italics

Author Response

We would like to the Reviewer for his/her comments and suggestions that helped us to improve our manuscript. Please find our responses below.

Comment 1: Title: Please replace metagenomics by metataxonomic.

Comments: Metataxonomics is a term proposed by Marchesi and Ravel and is defined as the high-throughput process used to characterize the entire microbiota and create a metataxonomic tree, which shows the relationships between all sequences obtained. “Metataxonomic analysis, because it relies on the amplification and sequencing of taxonomic marker genes, is not metagenomics”. Metagenomics refers to shotgun sequencing of DNA extracted from a sample

MARCHESI, Julian R. et RAVEL, Jacques. The vocabulary of microbiome research: a proposal. 2015.

Reply: We have now revised the title of the manuscript according to the Reviewer’s suggestion as follows: Unraveling the microbiota of natural black cv. Kalamata fermented olives through 16S and ITS metataxonomic analysis

Comment 2: Lines 176-177: Please specify the sequencing platform: Illumina MiSeq, Ion S5 XL or PAC BIO SEQUEL? 

Reply:

Results: The sequencing platform has been added in the revised manuscript (please see line 185 in the revised manuscript)

Comment 3: Figures 2, 3, 4 and 5 should be grouped in Figure 2a,b,c, and d. 

Reply: Figures 2, 3, 4 and 5 have been grouped (Figure 2 in the revised manuscript)

Comment 4: Figures 6, 8, 9, 10, 11, 12, and … the quality should be improved

Reply: The quality of all Figures has been improved

Comment 5: Figure 7 should be added in the supplemental material.

Reply: Figure 7 is now presented as Supplementary Figure 6 according to the Reviewer’s suggestion

Comment 6: Figures 15 and 17 X-axis, The name of microorganisms should be modifier. Capital letter, italics

Reply: The names in Figures 15 and 17 (in the revised manuscript Figures 11 and 13) have been corrected